# Revisiting Gaussian Neurons for Online Clustering with Unknown Number of Clusters

## Abstract

Despite the recent success of artificial neural networks, more biologically plausible learning methods may be needed to resolve the weaknesses of backpropagation trained models such as catastrophic forgetting and adversarial attacks. Although these weaknesses are not specifically addressed, a novel local learning rule is presented that performs online clustering with an upper limit on the number of clusters to be found rather than a fixed cluster count. Instead of using orthogonal weight or output activation constraints, activation sparsity is achieved by mutual repulsion of lateral Gaussian neurons ensuring that multiple neuron centers cannot occupy the same location in the input domain. An update method is also presented for adjusting the widths of the Gaussian neurons in cases where the data samples can be represented by means and variances. The algorithms were applied on the MNIST and CIFAR-10 datasets to create filters capturing the input patterns of pixel patches of various sizes. The experimental results[1] demonstrate stability in the learned parameters across a large number of training samples.

## 1 Introduction

Machine learning models trained through backpropagation have become widely popular in the last decade since AlexNet (Krizhevsky et al., 2012). Backpropagation, however, is not considered biologically plausible (Bengio et al., 2016), and the trained models are among other things susceptible to catastrophic forgetting (McCloskey & Cohen, 1989; Ratcliff, 1990) and adversarial attacks (Szegedy et al., 2014).

More biologically plausible methods would need to incorporate not only local learning rules and unsupervised learning, but also have properties such as sparser activations, capacity for distribution changes of the input, not prone to catastrophic forgetting, and work in an online setting, where samples are processed sequentially during learning one at a time.

One way to address distribution shifts in online clustering is to utilize overparameterized models that have additional parameters available to model input from subsequent unknown distributions. This will require that the parameters used to capture previous data be less likely to be adjusted, but model susceptibility to catastrophic forgetting would consequently be alleviated.

In this article, a novel learning rule is presented that utilizes neurons expressed with Gaussian functions. For a given neuron, the online method has an attraction term toward the current sample, and an inhibition term that ensures reduced overlap between the Gaussian neurons in the same layer to achieve activation sparsity. This makes it possible to have an overparameterized model with more neurons in a layer than is needed to represent the input. Some neurons will model previous input samples, while other neurons can adapt to new input from a possibly different distribution. While the inhibition term does not fully resolve the possibility of catastrophic forgetting, specialized rules may be added since the neurons representing already sampled data can be identified.

---

[1]The source code to reproduce the figures in this article is available at `https://gitlab.com/anonymous/gaussian-neurons-for-online-clustering`

Additionally, an update method for the neuron widths is presented, where the sample variances are measured from the data samples, and used to perform a constrained update of the neuron widths toward the sample variances. This can in certain conditions lead to a more robust learning rule whose results are less affected by the choice of initial neuron widths.

A limitation of the presented method is the use of isotropic Gaussian functions, employing scalar variances instead of covariance matrices. Moreover, Gaussian functions require additional computational resources and can be more numerically unstable compared to linear functions. Gaussian functions can, however, be approximated while retaining desired properties, for instance by using piecewise linear functions.

Finally, no claims are made that the proposed learning rule achieves better clustering results compared to previous work. On the other hand, the presented methods have a few interesting properties, as demonstrated in Section 4 such as examples of robustness to model disruption and distribution shift, which merit further studies toward more biologically plausible artificial neural networks that may be less susceptible to the weaknesses of current trained models.

## 2 Related Work

Clustering in an online setting has previously been studied extensively (Du, 2010), including for instance MacQueen's $k$-means (MacQueen et al., 1967), self-organizing maps (Kohonen, 1982), adaptive resonance theory (Carpenter & Grossberg, 1987), and neural gas (Fritzke et al., 1995). These algorithms typically implement a winner-take-all scheme, where only the winning neuron is adjusted with respect to a given sample.

Other methods extends Oja's learning rule (Oja, 1982) to achieve online clustering, such as White's competitive Hebbian learning rule (White, 1992) that adds a lateral inhibition term to the Oja's learning rule, and Decorrelated Hebbian Learning (Deco & Obradovic, 1995) where softmax normalized Gaussian functions are used for neuron activations. More recently, Pehlevan & Chklovskii (2014) presented an online clustering algorithm using symmetric non-negative matrix factorization based on the works of Ding et al. (2005); Kuang et al. (2012), and Krotov & Hopfield (2019) extended the Oja's learning rule with, moreover, a lateral inhibition term. The latter method was parallelized on GPUs in Grinberg et al. (2019) and Talloen et al. (2021). All of these methods, however, use a fixed number of clusters that the algorithms are predetermined to find.

Current artificial neuron models are overly simplified compared to cortical neurons (Beniaguev et al., 2021). In artificial neural networks, an activation function is usually used with a linear combination of input variables. On the other hand, clustering algorithms such as $k$-means and Gaussian mixture models use distance and multivariate Gaussian functions, respectively. The centroids or means of Gaussian functions capture more significantly both the angle and magnitude of the input vector, than that of linear combinations of input variables. Furthermore, variance and especially covariance provide additional flexibility for modeling the input. To model a convex region, for example, an artificial neural network using linear combination of the input variables will need two fully-connected layers each with a non-linear activation function, typically Rectified Linear Units since Glorot et al. (2011). The first layer will then represent hyperplanes in the input domain, and the second layer may combine these hyperplanes to convex regions. Gaussian functions, on the other hand, innately capture convex regions, potentially reducing the number of layers required for a given task. In terms of number of layers, a biological cortex cannot process signals sequentially as many times as the deeper artificial feedforward networks do, due to the relatively low firing rate in biological neurons (Wang et al., 2016). As a consequence, more biologically plausible models must perhaps be more shallow than is common in today's deep artificial networks.

Sparse representations can be an effective way to reduce the dimensionality of the input space, in addition to having the potential for more efficient use of resources. Methods relying on linear combinations of the input variables, such as sparse coding (Olshausen & Field, 1997), typically add a regularization term on the output activations. These algorithms, however, depend on reconstruction errors and are therefore not viable in certain unsupervised settings such as clustering. Another possibility is to regularize the linear coefficients directly as in competitive Hebbian learning (White, 1992) or independent component analysis

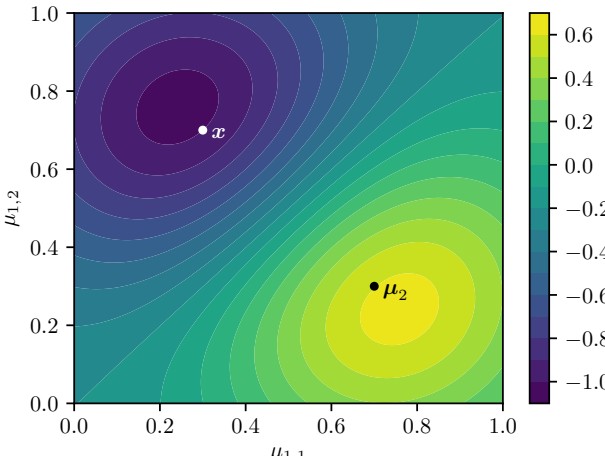

Figure 1: The cost function $F$ with $K = 2$ Gaussian neurons parameterized by the first neuron $\boldsymbol{\mu}_1$, where $\boldsymbol{x} = \begin{pmatrix} 0.3 & 0.7 \end{pmatrix}$, $\boldsymbol{\mu}_2 = \begin{pmatrix} 0.7 & 0.3 \end{pmatrix}$, $\sigma_1 = \sigma_2 = 0.2$, and $\lambda = \frac{1}{2}$. Minimizing $F$ with respect to $\boldsymbol{\mu}_1$ can be achieved by moving $\boldsymbol{\mu}_1$ closer to $\boldsymbol{x}$ and farther from $\boldsymbol{\mu}_2$. The minimum of $F$ is here at $\boldsymbol{\mu}_1 \approx \begin{pmatrix} 0.24 & 0.76 \end{pmatrix}$, although the presented learning rule will only make small changes to the cluster centers $\boldsymbol{\mu}_{1:K}$ for each $\boldsymbol{x}$. Furthermore, setting $\boldsymbol{\mu}_2 = \boldsymbol{x}$ would lead to a flat surface where $F = -1$, and in this case there would be no attraction or repulsion on $\boldsymbol{\mu}_1$.

(Comon, 1994), but these methods rely on orthogonal constraints, which limits the clustering method, i.e. the number of clusters found is generally less than or equal to the input dimensions, and two or more clusters cannot exist along a given vector from the origin. On the other hand, Deco & Obradovic (1995) made use of softmax normalized Gaussian neurons and regularization was achieved by penalizing overlapping Gaussian functions in the same layer. The neuron and cost function of this method, however, can result in learned cluster centers that are outside of the input domain, and these centers would then poorly represent the true cluster centers. Moreover, the proposed learning rule does not require a softmax function to produce the neuron outputs.

## 3 Learning Rule

Let $\boldsymbol{x} \in \mathbb{R}^D$ be an input vector, and $\boldsymbol{\mu}_i \in \mathbb{R}^D$ and $\sigma_i \in \mathbb{R}_{>0}$ be center and width, respectively, of the $i$-th Gaussian neuron in a layer, where $i \in \{1, \ldots, K\}$. The output of the $i$-th neuron is then defined as:

$$f_i(\boldsymbol{x}) = e^{-\|\boldsymbol{x} - \boldsymbol{\mu}_i\|_2^2 / \sigma_i}$$

where $\|\cdot\|_2$ denotes the $l_2$-norm. The objective of the learning rule is to minimize the cost function $E$ with respect to the centers $\boldsymbol{\mu}_{1:K}$:

$$E(\mathbb{X}; \boldsymbol{\mu}_1, \ldots, \boldsymbol{\mu}_K, \sigma_1, \ldots, \sigma_K) = \sum_{\boldsymbol{x} \in \mathbb{X}} \sum_{i=1}^{K} \left( -f_i(\boldsymbol{x}) + \lambda \sum_{j \neq i} f_j(\boldsymbol{\mu}_i) \right)$$

where $\mathbb{X} \in \mathbb{R}^{N \times D}$ is a potentially very large set of $N$ training examples that $\boldsymbol{x}$ is sampled from, $K$ is the number of Gaussian neurons, and $\lambda \in \mathbb{R}_{>0}$ controls the level of inhibition.

The first term of $E$ attracts the cluster centers $\boldsymbol{\mu}_{1:K}$ toward the input $\boldsymbol{x}$, while the second term repels the different cluster centers away from each other depending on the Gaussian locations and widths. Interestingly, setting $\lambda \geq \frac{1}{2}$ would nullify all attraction toward the given input if a cluster center already has the same position as the input and the Gaussian widths are equal. Lowering $\lambda$ and $\sigma_{1:K}$ has the same effect, that is

more input patterns can be learned, but smaller widths $\sigma_{1:K}$ may lead to insufficient attraction of the cluster centers toward the input.

In order to minimize $E$ in an online setting, the learning rule will update $\boldsymbol{\mu}_{1:K}$ by small steps in the opposite direction, and proportionally to the magnitude, of the gradient of the second sum of $E$:

$$F(\boldsymbol{x}; \boldsymbol{\mu}_1, \ldots, \boldsymbol{\mu}_K, \sigma_1, \ldots, \sigma_K) = \sum_{i=1}^{K} \left( -f_i(\boldsymbol{x}) + \lambda \sum_{j \neq i} f_j(\boldsymbol{\mu}_i) \right)$$

at input $\boldsymbol{x}$. See Figure 1 for an example surface of $F$ with two neurons. For each input $\boldsymbol{x}$, the cluster centers will be updated as follows:

$$\boldsymbol{\mu}_i = \boldsymbol{\mu}_i + \Delta\boldsymbol{\mu}_i = \boldsymbol{\mu}_i - \frac{1}{2}\eta_\mu \frac{\partial F}{\partial \boldsymbol{\mu}_i}$$

where $\eta_\mu \in \mathbb{R}_{>0}$ is the learning rate. The learning rule then becomes:

$$\Delta\boldsymbol{\mu}_i = \eta_\mu \left( \frac{f_i(\boldsymbol{x})}{\sigma_i}(\boldsymbol{x} - \boldsymbol{\mu}_i) - \lambda \sum_{j \neq i} \left( \frac{f_i(\boldsymbol{\mu}_j)}{\sigma_i} + \frac{f_j(\boldsymbol{\mu}_i)}{\sigma_j} \right)(\boldsymbol{\mu}_j - \boldsymbol{\mu}_i) \right) \tag{1}$$

If $\sigma_i = \sigma \ \forall i \in \{1, \ldots, K\}$, the learning rule can be simplified to:

$$\Delta\boldsymbol{\mu}_i = \frac{\eta_\mu}{\sigma} \left( f_i(\boldsymbol{x})(\boldsymbol{x} - \boldsymbol{\mu}_i) - 2\lambda \sum_{j \neq i} f_i(\boldsymbol{\mu}_j)(\boldsymbol{\mu}_j - \boldsymbol{\mu}_i) \right)$$

When applying the learning rule in Equation 1, some cluster centers may be repelled outside of the input domain. However, this is a beneficial property that can be utilized to identify learned cluster centers that correspond to patterns found from the input, as these centers will lie within or relatively close to the input domain.

The maximum distance a cluster center can be pushed outside of the input domain is constrained by the input domain itself, the parameters of Equation 1, and the number of clusters $K$. The maximum distance goes toward $\infty$ as the number of clusters approaches $\infty$, but considering $D = 1$, $K = 2$, $\lambda = \frac{1}{2}$, $\mu_2 = 0$, $\sigma_1 = \sigma_2 = \sigma$, and $f_i(x) \to 0 \ \forall i \in \{1, 2\} \ \forall x \in \mathbb{R}$, the maximum distance between $\mu_1$ and $\mu_2$ converges to the solution of the ordinary differential equation at $\mu_1(t)$ as $t \to \infty$:

$$\frac{\partial \mu_1}{\partial t} = -\frac{\partial F}{\partial \mu_1} = 2\frac{\mu_1}{\sigma}e^{-\mu_1^2/\sigma}, \quad \mu_1(0) > 0$$

Although in practice, the learning rate, the number of iterations, and the precision of the data types will limit the above distance.

### 3.1 Update Method for Neuron Widths

Having sparsity constraints given by Gaussian functions enables adjustments of the activation sparsity based on sample measures. The Gaussian widths $\sigma_{1:K}$ can be derived from the data $\mathbb{X}$ if a data sample is instead represented by a mean vector $\boldsymbol{x} = \boldsymbol{\mu}_x$ and a width scalar $\sigma_x$. This addition to the learning rule alters a neuron width $\sigma_i$ toward the given data sample width $\sigma_x$ depending on constraints such as how well the $i$-th Gaussian matches the data sample. Furthermore, care must be taken to avoid neuron functions collapsing into each other, i.e. resulting in several tight clusters near similar input patterns that have a relatively high probability of being sampled.

Let the measure of a sample be:

$$f_x(\boldsymbol{\mu}_i) = e^{-\|\boldsymbol{\mu}_i - \boldsymbol{\mu}_x\|_2^2 / \sigma_x}$$

and the update of the neuron widths be defined as:

$$\Delta\sigma_i = \eta_\sigma \max\left\{f_x(\boldsymbol{\mu}_i) - 2\lambda\sum_{j \neq i} f_j(\boldsymbol{\mu}_i), 0\right\} f_i(\boldsymbol{\mu}_x)(\sigma_x - \sigma_i) \tag{2}$$

where $\eta_\sigma \in \mathbb{R}_{>0}$ controls the rate of change, and $\lambda$ is the inhibition level used in Equation 1. The change of a neuron width $\sigma_i$ is thus relying on the neuron mean with respect to the sample measure inhibited by the other lateral neurons, and the sample mean with respect to the neuron function. The $max\{\cdot\}$ function ensures that the change of $\sigma_i$ is always 0 or toward $\sigma_x$.

Equations 1 and 2 can be applied independently, but adjusting means and widths interchangeably can be advantageous. For example, by enabling additional clusters to be found after the already identified clusters have had their widths reduced. Experiments employing both Equations 1 and 2 are presented in Subsection 4.3.

## 4    Results and Discussion

The presented learning rule was tested on image patches from the MNIST (Lecun et al., 1998) and CIFAR-10 (Krizhevsky, 2009) datasets. The $D$ pixel patch intensities were scaled to be from 0 to 1. No other preprocessing was applied to the images or image patches.

### 4.1    MNIST

$N$ uniformly random $D = 5 \times 5$ pixel patches were sampled from the training set containing $6 \times 10^6$ images of handwritten digits. Figure 2 shows the resulting centers, typically named filters in this context, of $K = 16$ clusters with Gaussian widths $\sigma_i = \sigma \; \forall i \in \{1, \ldots, K\}$, inhibition level $\lambda$, and learning rate $\eta_\mu = 10^{-1}$.

Some cluster centers have not changed significantly from training, while other filters represent learned filters that for instance can act as edge detection filters. The unchanged filters have typically been pushed outside of the input domain, and have a high cosine similarity to the initial randomized filters, as shown in Table 1 for the filters depicted in Subfigure 2a.

|  | $\mu_1$ | $\mu_2$ | $\boldsymbol{\mu}_3$ | $\boldsymbol{\mu}_4$ | $\boldsymbol{\mu}_5$ | $\boldsymbol{\mu}_6$ | $\mu_7$ | $\mu_8$ | $\mu_9$ | $\mu_{10}$ | $\mu_{11}$ | $\boldsymbol{\mu}_{12}$ | $\mu_{13}$ | $\mu_{14}$ | $\boldsymbol{\mu}_{15}$ | $\boldsymbol{\mu}_{16}$ |
|---|---|---|---|---|---|---|---|---|---|---|---|---|---|---|---|---|
| $d_i$ | 1.5 | 1.5 | **1.0** | **1.0** | **0.8** | **0.9** | 1.4 | 1.4 | 1.5 | 1.4 | 1.5 | **0.9** | 1.6 | 1.5 | **0.8** | **0.9** |
| $\cos\theta_i$ | 0.9 | 0.8 | **0.4** | **−0.1** | **0.8** | **0.7** | 0.9 | 0.9 | 0.9 | 0.9 | 0.9 | **0.7** | 0.9 | 0.9 | **0.8** | **0.6** |

Table 1: Scalar descriptors of the filters shown in Subfigure 2a, where $d_i = \|\boldsymbol{\mu}_i - \frac{1}{2}\|_2 / \sqrt{\frac{D}{4}}$, $\sqrt{\frac{D}{4}}$ is the maximum distance from a point in the input domain to the input domain centroid, $\cos\theta_i = \boldsymbol{\mu}_i \cdot \boldsymbol{\mu}_i^{(init)} / (\|\boldsymbol{\mu}_i\|_2 \|\boldsymbol{\mu}_i^{(init)}\|_2)$, and $\boldsymbol{\mu}_i^{(init)}$ is the $i$-th initial randomized filter before training. The learned filters, which for instance can act as edge detectors, are marked in bold.

The difference between the results shown in Subfigures 2a and 2b is that the latter experiment was run on 10 times as many data samples. Regardless, the filters shown in Subfigure 2b are similar to those in Subfigure 2a, demonstrating that the filters can be stable throughout a high number of iterations. Both experiments used the same random generator seed.

Additional patterns were found in the results shown in Subfigures 2c and 2d by lowering either the Gaussian width $\sigma$ or the inhibition level $\lambda$, respectively. 11 and 12 learned filters are present in Subfigures 2c and 2d compared to 8 filters in Subfigure 2b, with the same number of samples processed. Reducing the Gaussian

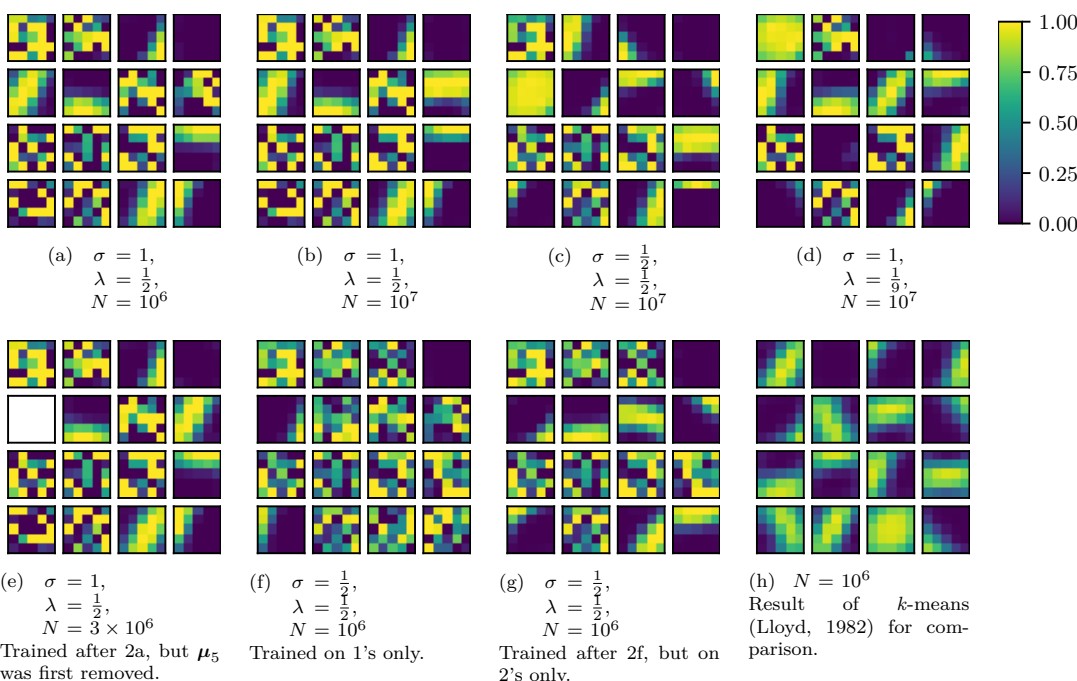

(a)  $\sigma = 1$,
$\lambda = \frac{1}{2}$,
$N = 10^6$

(b)  $\sigma = 1$,
$\lambda = \frac{1}{2}$,
$N = 10^7$

(c)  $\sigma = \frac{1}{2}$,
$\lambda = \frac{1}{2}$,
$N = 10^7$

(d)  $\sigma = 1$,
$\lambda = \frac{1}{9}$,
$N = 10^7$

(e)  $\sigma = 1$,
$\lambda = \frac{1}{2}$,
$N = 3 \times 10^6$
Trained after 2a, but $\boldsymbol{\mu}_5$ was first removed.

(f)  $\sigma = \frac{1}{2}$,
$\lambda = \frac{1}{2}$,
$N = 10^6$
Trained on 1's only.

(g)  $\sigma = \frac{1}{2}$,
$\lambda = \frac{1}{2}$,
$N = 10^6$
Trained after 2f, but on 2's only.

(h)  $N = 10^6$
Result of $k$-means (Lloyd, 1982) for comparison.

Figure 2: The resulting centers or filters from applying the presented learning rule (2a to 2g) with $K = 16$ neurons, $N$ $5 \times 5$ pixel patches uniformly sampled from the MNIST training dataset, Gaussian widths $\sigma_i = \sigma \ \forall i \in \{1, \ldots, K\}$, inhibition level $\lambda$, and learning rate $\eta_\mu = 10^{-1}$. The learned filters can for instance act as edge detectors, while the remaining filters were pushed outside of the input domain and had a high cosine similarity to the initial filter values prior to training. The filters are enumerated left to right, top to bottom.

widths can, however, lead to fewer identified patterns since the pull toward some data samples may be insufficient. In addition, if the inhibition level is too low, filters can collapse into each other, forming several identical Gaussian centers. For example, 2 equal Gaussian centers $\boldsymbol{\mu}_4 = \boldsymbol{\mu}_{13} \approx (0, \ldots, 0)$ were found when the experiment shown in Subfigure 2d was run with the inhibition level lowered to $\lambda = 10^{-1}$.

In the result shown in Subfigure 2e, the learned filters remain approximately the same as in Subfigure 2a, despite the fact that the learned filter $\boldsymbol{\mu}_5$ was removed prior to training on additional $N = 3 \times 10^6$ data samples. The pattern learned by filter $\boldsymbol{\mu}_5$ in Subfigure 2a was then relearned by filter $\boldsymbol{\mu}_8$ in Subfigure 2e. This result demonstrates potential stability of the learning rule as a pattern can be relearned after disruption while the other learned filters can remain largely unaltered.

Subfigure 2f shows the result of training solely on images depicting the number 1, and this model was subsequently used as a starting point to produce the result shown in Subfigure 2g, where only images corresponding to the number 2 were used during training. Even though there was a distribution change of the inputs, the three filters found in Subfigure 2f are largely similar to the corresponding filters in Subfigure 2g, while additional filters were learned as well. The additional filters mainly represents patterns found exclusively in the latter dataset.

For comparison, Subfigure 2h shows the resulting filters from the $k$-means algorithm (Lloyd, 1982) with the cluster count set to 16. The proposed learning rule will most likely not learn as many filters from the MNIST dataset given the relatively small pixel patch size without setting the Gaussian width $\sigma$ or inhibition level $\lambda$ too low.

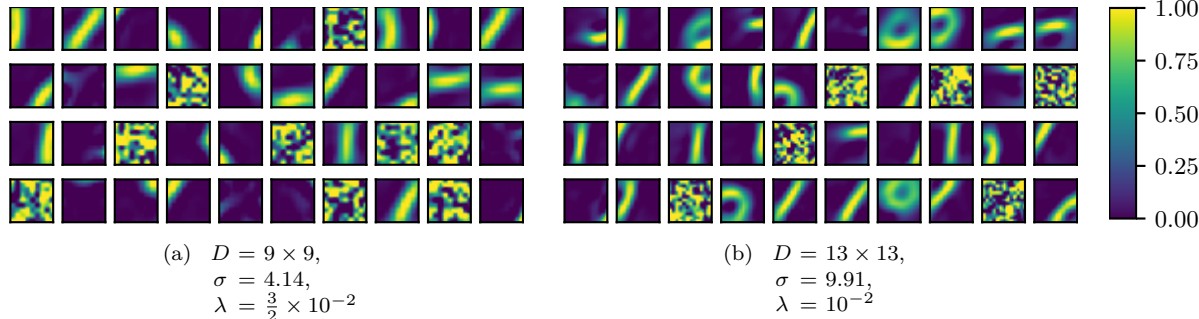

(a) $D = 9 \times 9$,
$\quad \sigma = 4.14$,
$\quad \lambda = \frac{3}{2} \times 10^{-2}$

(b) $D = 13 \times 13$,
$\quad \sigma = 9.91$,
$\quad \lambda = 10^{-2}$

Figure 3: The results of employing the learning rule on larger image patches. In both experiments, the number of samples was $N = 10^7$, the Gaussian widths set to $\sigma_i = \sigma \; \forall i \in \{1, \ldots, K\}$, and the learning rate equal to $\eta_\mu = 10^{-1}$. More rounded filters were learned compared to those in Figure 2, but the Gaussian widths had to be increased in order to better capture the patterns in the larger input domain.

### 4.1.1 Larger Pixel Patches

Experiment results with $N = 10^7$ samples of $D = 9 \times 9$ and $D = 13 \times 13$ pixel patches are shown in Figure 3. Similar to Figure 2, the Gaussian widths were set to $\sigma_i = \sigma \; \forall i \in \{1, \ldots, K\}$. More rounded filters were learned, especially those shown in Subfigure 3b, which can be used to detect parts of particularly the handwritten digits 0, 2, 3, 5, 6, 8, and 9.

As the number of input dimensions was increased, leading to a larger input domain, the inhibition level was decreased and the Gaussian widths were increased to better capture the patterns of the input samples. To make the results more comparable with, for instance, the results in Subfigure 2d, the width $\sigma$ was calculated such that two neurons placed on opposite positions of the input domains had equal repulsion, measured by the $l_2$-norm normalized by the number of dimensions, independent of input samples:

$$\frac{1}{D}\|\Delta\boldsymbol{\mu}_1\|_2 = \frac{1}{\tilde{D}}\|\Delta\tilde{\boldsymbol{\mu}}_1\|_2$$

$$\frac{1}{D}\left\|\frac{\lambda\boldsymbol{\mu}_1}{\sigma^2}e^{-\|\boldsymbol{\mu}_1\|_2^2/\sigma}\right\|_2 = \frac{1}{\tilde{D}}\left\|\frac{\tilde{\lambda}\tilde{\boldsymbol{\mu}}_1}{\tilde{\sigma}^2}e^{-\|\tilde{\boldsymbol{\mu}}_1\|_2^2/\tilde{\sigma}}\right\|_2 \quad \text{s.t.} \quad \begin{array}{c} \sigma > 0 \\ \frac{\partial}{\partial\sigma}\frac{\lambda}{\sqrt{D}\sigma^2}e^{-D/\sigma} > 0 \end{array} \tag{3}$$

$$\frac{\lambda}{\sqrt{D}\sigma^2}e^{-D/\sigma} = \frac{\tilde{\lambda}}{\sqrt{\tilde{D}}\tilde{\sigma}^2}e^{-\tilde{D}/\tilde{\sigma}}$$

where $K = 2$, $\boldsymbol{\mu}_1 = (1, \ldots, 1)$, $\boldsymbol{\mu}_2 = (0, \ldots, 0)$, $\boldsymbol{\mu}_i \in \mathbb{R}^D$, $f_1(\boldsymbol{x}) \to 0 \; \forall \boldsymbol{x} \in \mathbb{R}^D$, $\tilde{\boldsymbol{\mu}}_1 = (1, \ldots, 1)$, $\tilde{\boldsymbol{\mu}}_2 = (0, \ldots, 0)$, $\tilde{\boldsymbol{\mu}}_i \in \mathbb{R}^{\tilde{D}}$, and $\tilde{f}_1(\tilde{\boldsymbol{x}}) \to 0 \; \forall \tilde{\boldsymbol{x}} \in \mathbb{R}^{\tilde{D}}$. The tilde symbol denotes the vectors and scalars used in this case to produce the result presented in Subfigure 2d, and the learning rate $\eta_\sigma$ was equal in both $\Delta\boldsymbol{\mu}_1$ and $\Delta\tilde{\boldsymbol{\mu}}_1$.

From Equation 3 then with $\tilde{D} = 5 \times 5$, $\tilde{\sigma} = 1$, $\tilde{\lambda} = \frac{1}{9}$, the neuron widths for computing the results shown in the Subfigures 3a and 3b were approximated to be $\sigma = 4.14$ and $\sigma = 9.91$, respectively. The inhibition level was made slightly different in order to find roughly the same number of filters, where $\lambda = \frac{3}{2} \times 10^{-2}$ was used to process $D = 9 \times 9$ pixel patches and $\lambda = 10^{-2}$ to process $D = 13 \times 13$ pixel patches.

### 4.2 CIFAR-10

The learning rule was also applied on the CIFAR-10 dataset (Krizhevsky, 2009), where $N$ uniformly random $D = 5 \times 5 \times 3$ pixel patches were sampled from the training set containing $6 \times 10^6$ natural images, evenly portraying airplanes, cars, birds, cats, deer, dogs, frogs, horses, ships, and trucks. Figure 4 shows the resulting filters of $K = 50$ clusters with Gaussian widths $\sigma_i = \sigma = 1.75 \; \forall i \in \{1, \ldots, K\}$, inhibition level $\lambda = 10^{-2}$, and learning rate $\eta_\mu = 10^{-1}$.

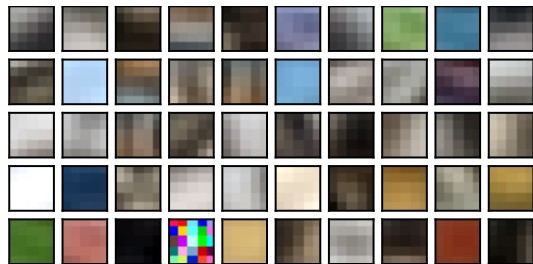

Figure 4: The resulting filters from training on $D = 5 \times 5 \times 3$ pixel patches from the CIFAR-10 dataset containing natural images, with $N = 10^7$ number of samples, Gaussian widths $\sigma_i = 1.75 \; \forall i \in \{1, \ldots, K\}$, inhibition level $\lambda = 10^{-2}$, and learning rate $\eta_\mu = 10^{-1}$. Some of the edge detecting filters are color independent, while some filters can be used to detect specific color patches such as sky and grass.

Some of the learned filters can be used to detect specific colored regions such as sky and grass, while other filters can act as edge detectors. Some of the edge detecting filters are gray and thus color independent. All filters except one learned patterns from the pixel patch samples.

## 4.3 Neuron Width Update

Both Equations 1 and 2 were employed to produce the results shown in Figure 5, using $N = 10^7$ data samples and learning rate set to $\eta_\mu = \eta_\sigma = 10^{-1}$. In Subfigures 5c to 5f, the data sample mean $\boldsymbol{x} = \boldsymbol{\mu}_x = \frac{1}{9}\sum_{i=1}^{9} \boldsymbol{s}_i$ and width $\sigma_x = 2\text{Var}[\boldsymbol{s}_{1:9}] = \frac{2}{9}\sum_{i=1}^{9}(\boldsymbol{s}_i - \boldsymbol{\mu}_x)^2$ were measured from 9 neighboring $5 \times 5$ image patches $\boldsymbol{s}_{1:9}$ after adding normal distributed noise with 0 mean, and 0.1 (Subfigures 5c to 5e) or 0.15 (Subfigure 5f) standard deviation, to each pixel value. Neighboring image patches can share similar patterns, and noise was added as a constraint to keep the Gaussian widths from collapsing toward 0, for example, when sampling homogeneous image regions.

The neuron means $\boldsymbol{\mu}_{1:K}$ and widths $\sigma_{1:K}$ were updated in alternate succession, that is for each data sample, the neuron means were first updated using Equation 1, and then Equation 2 was used to update the neuron widths.

Subfigures 5c to 5e show similar $K = 9$ filters even though the initial Gaussian widths $\sigma_i^{(init)} = \sigma^{(init)} \; \forall i \in \{1, \ldots, K\}$ were set to $\sigma^{(init)} = 6$, $\sigma^{(init)} = 10$, and $\sigma^{(init)} = 14$, respectively. For comparison, Subfigures 5a and 5b show the resulting filters without neuron width updates where the Gaussian widths $\sigma_i = \sigma \; \forall i \in \{1, \ldots, K\}$ were set to, in the same order, $\sigma = 4$ and $\sigma = 14$. The inhibition level was $\lambda = \frac{1}{2}$ for all Subfigures 5a to 5e.

The final Gaussian widths of the filters shown in Subfigure 5d were:

$$\sigma_{1:K} = \left(10, 9.5, \boldsymbol{4.6}, \boldsymbol{0.6}, \boldsymbol{5}, 10, 10, \boldsymbol{4.6}, 10\right)$$

and the learned filters can in this case be distinguished from the filters that have not learned any patterns by comparing the final widths to the initial widths of the Gaussians.

Lastly, Subfigure 5f shows the result of applying the learning rule and width update method on data samples from the CIFAR-10 dataset with initial Gaussian width $\sigma^{(init)} = 10$ and inhibition level $\lambda = 10^{-2}$. Almost the same number of filters were learned compared to those learned in Figure 4, although the initial widths were larger and each data sample was extracted from 9 neighboring $5 \times 5 \times 3$ pixel patches. By using Equation 2, however, the Gaussian widths of the learned filters were reduced and thus made room for additional Gaussians within the data domain.

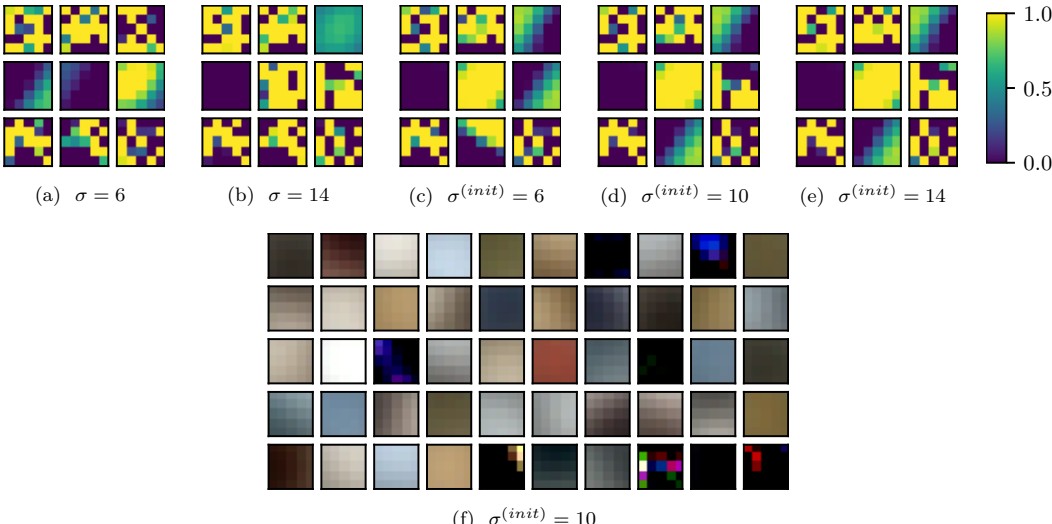

(a) $\sigma = 6$    (b) $\sigma = 14$    (c) $\sigma^{(init)} = 6$    (d) $\sigma^{(init)} = 10$    (e) $\sigma^{(init)} = 14$

(f) $\sigma^{(init)} = 10$

Figure 5: The results of employing both Equation 1 and 2 on data sample measures from the MNIST and CIFAR-10 datasets. Subfigures 5c to 5e show similar results even though the initial neuron widths were different. On the other hand 3 and 2 filters were learned when the neuron widths were fixed to $\sigma = 6$ and $\sigma = 14$ as shown in Subfigures 5a and 5b, respectively. The result from Figure 4 was reproduced in Subfigure 5f although with varying Gaussian widths. All results were produced using $N = 10^7$ data samples and learning rates set to $\eta_\mu = \eta_\sigma = 10^{-1}$.

## 5 Conclusion and Future Work

A novel unsupervised learning rule based on Gaussian functions is presented that can perform online clustering without needing to specify the number of clusters prior to training. The local learning rule is arguably more biologically plausible compared to model optimization through backpropagation, and the results demonstrate stability in the learned parameters during training. Furthermore, an update method for the widths of the Gaussian functions is presented, which can reduce the dependence on finely tuned hyperparameters.

Opportunities for future work include investigations of the presented methods in the following directions: (1) improve the computation speed through CPU or GPU parallelization, (2) use anisotropic Gaussian functions, (3) train with input data other than image patches, (4) layer-wise optimization of deeper model architectures, (5) compare trained model susceptibility to adversarial attacks against models optimized through backpropagation, (6) sample mean and variance from subsequent video frames instead of neighboring image patches, and (7) repurposing the Gaussian functions as Gaussian dendrites, and linearly combine similar dendritic functions into artificial neurons to achieve greater representational power of the artificial neurons.

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
