# OpenReview forum: "Revisiting Gaussian Neurons for Online Clustering with Unknown Number of Clusters"
_TMLR — Rejected by TMLR_

### Review · Reviewer_dUS4 · 2022-06-27

**Summary Of Contributions:**

The authors discuss an algorithm for online clustering using Gaussian neurons and test their algorithm on MNIST and CIFAR-10.

**Broader Impact Concerns:**

Not applicable.

**Requested Changes:**

I would suggest that the authors consider the take-away points of this paper and do a major revision focusing on these points. If the point is that the proposed algorithm improves performance under distribution shift / catastrophic forgetting, they should focus on this and compare with other algorithms aimed at this. If the point is that this is an efficient local clustering algorithm, the authors should focus on the efficiency and the shape of the network implementing this. Finally, if the point is that this is a local algorithm that can be biologically plausible and therefore relevant for the brain, the authors should be explicit about this.

In all cases, the paper should cite the relevant literature and perform comparison with prior work.

**Strengths And Weaknesses:**

Unfortunately the goal of the paper is not clear and the statements are often inaccurate or unjustified.

The authors also do not cite a very relevant prior work performing arguably the same type of classification in an online setting [1]. Since this paper published in 2017 is written with very similar functional goals (implementing k-means in and online and local way with gaussian neurons), it is crucial for the authors to cite and differentiate their work from [1] if they want to get published.

Here are some of the inaccurate/unjustified/confusing remarks:

1. Upper limit instead of fixed cluster count. Usually most clustering algorithms (e.g. k-means) work with an upper limit of the cluster count and can end up with empty clusters. This is in general treated as an undesirable feature since these algorithms do not claim that the number of non-empty clusters that are found is actually the correct number of clusters. I do not see how the proposed method is different from other clustering algorithms in this regard.

2. Cited methods use linear combinations of input variables. This is very confusing. The author's proposed method is based on a linear distance to some centroid (just like k-means), so it seems like a linear clustering algorithm to me. In comparison, the online clustering algorithm of the cited Pehlevan & Chkhlovskii (2014) uses non-negative neurons and is truly non-linear.

3. Regularizing output activations is not viable in an online setting. Why? This can be done using stochastic gradient descent.

4. Arguments about distribution shift and forgetting. The authors provide no evidence regarding this. The hand-wavy arguments are not very convincing.

5. Local learning rules. In Eq. 1, the update rule for mu_i needs the centroid of mu_j. How is this a local learning rule. Perhaps the authors should specify the shape of the network that they claim implements this learning rule locally.



[1] C. Pehlevan, A. Genkin and D. B. Chklovskii, "A clustering neural network model of insect olfaction," 2017 51st Asilomar Conference on Signals, Systems, and Computers, 2017, pp. 593-600, doi: 10.1109/ACSSC.2017.8335410.

---

> ### Author Response · Authors · 2022-07-06
> **Response to reviewer dUS4**
>
> Thank you for your careful reading of our work. We have uploaded an updated version of the manuscript based on your feedback.
>
> > Unfortunately the goal of the paper is not clear and the statements are often inaccurate or unjustified.
> >
> > The authors also do not cite a very relevant prior work performing arguably the same type of classification in an online setting [1]. Since this paper published in 2017 is written with very similar functional goals (implementing k-means in and online and local way with gaussian neurons), it is crucial for the authors to cite and differentiate their work from [1] if they want to get published.
> >
> > Here are some of the inaccurate/unjustified/confusing remarks:
> >
> > 1. Upper limit instead of fixed cluster count. Usually most clustering algorithms (e.g. k-means) work with an upper limit of the cluster count and can end up with empty clusters. This is in general treated as an undesirable feature since these algorithms do not claim that the number of non-empty clusters that are found is actually the correct number of clusters. I do not see how the proposed method is different from other clustering algorithms in this regard.
>
> The presented learning rule has many differences compared to k-means algorithms, for instance in our work, the cost function decreases the terms with increasing distances between cluster centers and data points, but the terms are never set to 0. These terms together with the inhibition terms were built around the premise that the number of clusters are overestimated to be able to model future samples from different and unseen distributions, with the possibility of retaining the previously learned centers. Furthermore, the cluster regions can be hyperspherical with different radii, with the possibility for, although only mentioned in the paper, hyperellipsoidally shaped clusters. Finally, related to the work in [1] and contrary to our work, both the classic online k-means algorithm and the method presented in [1] rely on a winner-take-all scheme, which [2], for example, states to be "nonbiological". The online k-means methods also need to track the number of samples assigned to each winning cluster.
>
> > 2. Cited methods use linear combinations of input variables. This is very confusing. The author's proposed method is based on a linear distance to some centroid (just like k-means), so it seems like a linear clustering algorithm to me. In comparison, the online clustering algorithm of the cited Pehlevan & Chkhlovskii (2014) uses non-negative neurons and is truly non-linear.
>
> We agree that this statement in the article is problematic, and have removed it in the updated version of the manuscript.
>
> > 3. Regularizing output activations is not viable in an online setting. Why? This can be done using stochastic gradient descent.
>
> This statement was made in the context of unsupervised learning, but it is still disputable and we have rewritten the sentence in the updated version of the manuscript.
>
> > 4. Arguments about distribution shift and forgetting. The authors provide no evidence regarding this. The hand-wavy arguments are not very convincing.
>
> We bring up distribution shift and catastrophic forgetting as motivation for working on backpropagation alternatives, but make no claims of having solved these challenges. However, we do demonstrate that the proposed learning rule can in certain conditions handle distribution shift and avoid catastrophic forgetting in the experiments shown in Subfigure 2f-g, and indirectly in Subfigure 2e. That said, we added a paragraph at the end of the Introduction stating that no claims are made that the proposed learning rule is better than previous methods, but the presented methods instead have interesting properties that merit further studies.
>
> > 5. Local learning rules. In Eq. 1, the update rule for mu_i needs the centroid of mu_j. How is this a local learning rule. Perhaps the authors should specify the shape of the network that they claim implements this learning rule locally.
>
> This depends on the definition of "local" in this context. We are using "local" as in [2] and [3], which includes lateral neurons. Our article states several times that the learning rule is dependent on lateral neurons, for instance "activation sparsity is achieved by mutual repulsion of lateral Gaussian neurons" and "an inhibition term that ensures reduced overlap between the Gaussian neurons in the same layer".
>
> [1] C. Pehlevan, A. Genkin and D. B. Chklovskii, "A clustering neural network model of insect olfaction", 2017 51st Asilomar Conference on Signals, Systems, and Computers, 2017, pp. 593-600, doi: 10.1109/ACSSC.2017.8335410.
>
> [2] G. Deco and D. Obradovic. "Decorrelated hebbian learning for clustering and function approximation". Neural Computation, 7(2):338–348, 1995. doi: 10.1162/neco.1995.7.2.338.
>
> [3] R. H White. "Competitive hebbian learning: Algorithm and demonstrations", Neural Networks, 5(2):261–275, 1992. doi: 10.1016/s0893-6080(05)80024-3.

---

### Review · Reviewer_MxzB · 2022-06-28

**Summary Of Contributions:**

This paper proposes a local learning rule that utilizes neurons expressed with Gaussian functions for performing online clustering without a prior number of clusters fixed. The means of Gaussian neurons are updated with an attraction term toward the current sample, and an inhibition term that ensures reduced overlap between the Gaussian neurons in the same layer. It also presents an update method for the variances of Gaussian neurons, to obtain a more robust learning rule whose results are less affected by the choice of initial neuron widths. The mechanisms of the proposed learning rule are validated by well-designed experiments on MNIST and CIFAR-10 datasets.

**Broader Impact Concerns:**

No impact concerns.

**Requested Changes:**

(1)	I personally think this paper can be accepted if it provides further statements for multiple-layers networks/systems with empirical evidences, like the current results for one-layer model presented in this paper.

(2)	It is not ready to understand this sentence, “Interestingly, setting λ ≥1/2  would nullify all attraction toward the given input if a cluster center already has the same position as the input and the Gaussian widths are equal.” It is better to provide more evidences. That is, if this argument is shown in previous related work, this paper should cite it. Otherwise, this paper should provide more clarification for this sentence. E.g., combining the update rule (Eqn.1) to illustrate it, and even provide a proof.



**Strengths And Weaknesses:**

My evaluation focuses on that: 1)  whether the claims made in the submission supported by accurate, convincing, and clear evidence; 2)  whether audience be interested in the findings of this paper. I cannot well judge the novelty and significance of this paper, compared to previous one, since I am not an expert relating to biological neural networks and online clustering.

##Strengths:

+ This paper provides some interesting findings. For example, the proposed unsupervised methods can well learn the (low-level) patterns, like the supervised neural networks do.
+ This paper provides good evidence to support its most claims, particularly using well designed toy examples/experiments to clarify the motivations and insights behind.
+ The presentation of this paper is very clear and I am enjoying in reading this paper.


##Weaknesses:

+  My main concern for this paper is its contributions are not well matched with the areas of (biological or artificial) neural networks, while this paper aims to claim. This paper starts the motivations from the problems of neural networks using back-propagation training (e.g., susceptible to catastrophic forgetting and adversarial attacks) and term the proposed methods as Gaussian `neurons’. However, the current work is simply for a one-layer model, not for multiple-layer model (that is, stacking multiple-layers Gaussian neurons with training algorithm provided). Without investigating a multiple-layer model, this paper just proposes an online clustering method, whose related work and the corresponding baselines (if needed for empirical comparison) are totally different to the ones described in this paper, and should pay more discussions on the widely investigated online clustering problem.
Even though I noted this paper provides a list for further work (including the layer-wise optimization of deeper model architectures), It should at least include how to adapt the proposed method for multiple-layer model architectures for obtaining an accept, as to me.

+ I think this paper should also provide some experiments to show how well the proposed unsupervised learning rules over the previous exist work, Either using the unsupervised evaluation or transferring to supervised learning.


Other minors:
(1) It is not clear in the sentence “The first layer will then represent hyperplanes in the input domain, and the second layer may combine these hyperplanes to convex regions.” How the second layer combine these hyperplanes to convex regions?
(2) “a inhibition” should be “an inhibition”

---

> ### Author Response · Authors · 2022-07-06
> **Response to reviewer MxzB**
>
> Thank you for your feedback and your careful reading of our work. Below we provide responses to your comments and questions. We have also uploaded an updated version of the manuscript based on your feedback.
>
> > My main concern for this paper is its contributions are not well matched with the areas of (biological or artificial) neural networks, while this paper aims to claim. This paper starts the motivations from the problems of neural networks using back-propagation training (e.g., susceptible to catastrophic forgetting and adversarial attacks) and term the proposed methods as Gaussian `neurons’. However, the current work is simply for a one-layer model, not for multiple-layer model (that is, stacking multiple-layers Gaussian neurons with training algorithm provided). Without investigating a multiple-layer model, this paper just proposes an online clustering method, whose related work and the corresponding baselines (if needed for empirical comparison) are totally different to the ones described in this paper, and should pay more discussions on the widely investigated online clustering problem. Even though I noted this paper provides a list for further work (including the layer-wise optimization of deeper model architectures), It should at least include how to adapt the proposed method for multiple-layer model architectures for obtaining an accept, as to me.
>
> Model architecture is beyond the scope of this article as the focus is on the new learning rule, and we do not claim that the method can advantageously be used as a replacement for backpropagation training as of now. We could have demonstrated a trained Convolutional Neural Network architecture with a few layers, but we are not confident that that is the best direction for future research employing the presented learning rule, and the trained model would likely underperform compared to a backpropagation trained model. One immediate challenge with a multilayer architecture is that the signal between the layers might need to be strengtened in some way in order for the next layer to be able to model relevant input patterns. This might, however, be overcome by repurposing the gaussian neurons as gaussian dendrites instead as mentioned in the Conclusion and Future Work section.
>
> > I think this paper should also provide some experiments to show how well the proposed unsupervised learning rules over the previous exist work, Either using the unsupervised evaluation or transferring to supervised learning.
>
> We do not claim that the learning rule is better than existing methods, but it has some interesting properties that we believe merit further research. A paragraph was added at the end of the Introduction to make this clearer.
>
> > It is not clear in the sentence “The first layer will then represent hyperplanes in the input domain, and the second layer may combine these hyperplanes to convex regions.” How the second layer combine these hyperplanes to convex regions?
>
> We corrected the related sentences in the updated version such that it is clear that the activation function is part of a layer. Regarding the combination of hyperplanes, we refer to, for example, the solution of a xor-like problem such as https://playground.tensorflow.org/#activation=relu&batchSize=10&dataset=circle&regDataset=reg-plane&learningRate=0.03&regularizationRate=0&noise=0&networkShape=3&seed=0.43897&showTestData=false&discretize=false&percTrainData=50&x=true&y=true&xTimesY=false&xSquared=false&ySquared=false&cosX=false&sinX=false&cosY=false&sinY=false&collectStats=false&problem=classification&initZero=false&hideText=false, where the hidden layer represents the hyperplanes and the output layer combines these hyperplanes into one convex region in the input domain. To see the convex region you might have to restart the training a few times such that the training loss becomes close to zero.
>
> > “a inhibition” should be “an inhibition”
>
> This has been corrected in the updated version of the manuscript. Thank you again.
>
> > It is not ready to understand this sentence, “Interestingly, setting λ ≥1/2 would nullify all attraction toward the given input if a cluster center already has the same position as the input and the Gaussian widths are equal.” It is better to provide more evidences. That is, if this argument is shown in previous related work, this paper should cite it. Otherwise, this paper should provide more clarification for this sentence. E.g., combining the update rule (Eqn.1) to illustrate it, and even provide a proof.
>
> It is perhaps most easily seen in the equation following Equation 1, where the widths are equal, that setting λ = 1/2 could nullify the positive term. The Figure 1 text also mentions that the cost function becomes F = -1, that is a flat surface, when λ = 1/2 and mu_2 = x, and this can be reproduced with the accompanying source code for Figure 1.

---

### Review · Reviewer_Vx3w · 2022-07-05

**Summary Of Contributions:**

The paper presents a novel clustering algorithm, based on gaussian neurons, by introducing a local learning rule. As opposed to previous methods, this clustering algorithm does not require a number of clusters and instead can use simply an upper limit on the number of clusters. The paper then analysis how such clustering looks on MNIST and CIFAR-10, analyzing the resulting filters.


**Broader Impact Concerns:**

Not relevant for this submission

**Requested Changes:**

Writing: the introduction section is a bit disorganized -- the separation between past results and the current work is a bit unclear. A possible solution is to divide this section into 2 parts, "previous work" and "introduction", clearly stating what was known by previous art and what is new in this work. This point is not as critical to acceptance as the others, but can significantly strengthen the work.

Motivation: The claims made about the motivation of the suggested algorithm (biologically plausible / avoid catastrophic forgetting / withstand adversarial attacks) should either be removed, or better explained in the text. For each such claim, I would expect some empiricalthatretical evidence that shows why the suggested algorithm is better than the existing methods. This point is critical.

Experimental design - This is the most critical point. The results currently show visualizations of filters when running the suggested algorithm on MNIST and CIFAR-10. However, the properties of the algorithm (as opposed to other clustering algorithms) are unclear. I would expect a more comparative approach: how the suggested algorithm is performing as opposed to other clustering algorithms? which data is more suited to work on? why the shown filters are good? what could be expected if I will run the algorithm on new data?


**Strengths And Weaknesses:**

Strengths:
Novelty -- new clustering algorithm
Replacing the number of clusters with an upper bound result in a more generalized algorithm

Weaknesses:
Motivation -- the paper claims that "more biologically plausible learning methods may be needed to resolve the weaknesses of backpropagation trained models such as catastrophic forgetting and adversarial attacks". Yet, I couldn't find in the paper any claims on why the suggested algorithm is more biologically plausible and how it handles weaknesses such as catastrophic forgetting and adversarial attacks. Moreover, it is unclear from the paper what are the advantages of the proposed algorithm over existing methods.

Experimental design -- the paper show visualizations of the learned filters when training the suggested model on MNIST and CIFAR-10. It is unclear to me what could be learned from these results -- are the results of the clustering algorithm on the suggested data are good or bad? what metrics did you use to determine it? How did the proposed algorithm perform in comparison to other clustering algorithms? for which types of data is it more suited?
At the current state, the results feel a bit too anecdotical, and I'm struggling to see how what are the take-home messages for other problems.

---

> ### Author Response · Authors · 2022-07-06
> **Response to reviewer Vx3w**
>
> Thank you for your feedback and your careful reading of our work. Below we provide responses to your comments and questions. We have also uploaded an updated version of the manuscript based on your feedback.
>
> > Weaknesses: Motivation -- the paper claims that "more biologically plausible learning methods may be needed to resolve the weaknesses of backpropagation trained models such as catastrophic forgetting and adversarial attacks". Yet, I couldn't find in the paper any claims on why the suggested algorithm is more biologically plausible and how it handles weaknesses such as catastrophic forgetting and adversarial attacks. Moreover, it is unclear from the paper what are the advantages of the proposed algorithm over existing methods.
> >
> > Experimental design -- the paper show visualizations of the learned filters when training the suggested model on MNIST and CIFAR-10. It is unclear to me what could be learned from these results -- are the results of the clustering algorithm on the suggested data are good or bad? what metrics did you use to determine it? How did the proposed algorithm perform in comparison to other clustering algorithms? for which types of data is it more suited? At the current state, the results feel a bit too anecdotical, and I'm struggling to see how what are the take-home messages for other problems.
>
> We bring up distribution shift and catastrophic forgetting as motivation for working on backpropagation alternatives, but make no claims of having solved these challenges. However, we do demonstrate that the proposed learning rule can in certain conditions handle distribution shift and avoid catastrophic forgetting in the experiments shown in Subfigure 2f-g, and indirectly in Subfigure 2e. Furthermore, the presented algorithm does not require, for instance, a winner-take-all scheme, which [1], for example, states to be "nonbiological". That said, we added a paragraph at the end of the Introduction stating that no claims are made that the proposed learning rule is better than previous methods, but the presented methods instead have interesting properties that merit further studies.
>
> > Writing: the introduction section is a bit disorganized -- the separation between past results and the current work is a bit unclear. A possible solution is to divide this section into 2 parts, "previous work" and "introduction", clearly stating what was known by previous art and what is new in this work. This point is not as critical to acceptance as the others, but can significantly strengthen the work.
>
> We will consider dividing the Introduction section into two parts. This has also been considered by the authors previously.
>
> > Motivation: The claims made about the motivation of the suggested algorithm (biologically plausible / avoid catastrophic forgetting / withstand adversarial attacks) should either be removed, or better explained in the text. For each such claim, I would expect some empiricalthatretical evidence that shows why the suggested algorithm is better than the existing methods. This point is critical.
>
> As mentioned before, we bring up distribution shift and catastrophic forgetting as motivation for working on backpropagation alternatives. Adversarial attack vulnerability is not considered to be alleviated by the learning rule directly, but is mentioned in the Conclusion and Future Work section as a possible research direction. However, this would likely require a deeper model architecture, which is beyond the scope of this article.
>
> > Experimental design - This is the most critical point. The results currently show visualizations of filters when running the suggested algorithm on MNIST and CIFAR-10. However, the properties of the algorithm (as opposed to other clustering algorithms) are unclear. I would expect a more comparative approach: how the suggested algorithm is performing as opposed to other clustering algorithms? which data is more suited to work on? why the shown filters are good? what could be expected if I will run the algorithm on new data?
>
> In addition to the above replies, the submitted manuscript presents basic research toward more biologically plausible training of artificial neural networks, that is applied use of the algorithms is not considered at this stage. Furthermore, the presented methods are, to our knowledge, quite different from previous work. The only article we could find that was somewhat similar was [1], but their algorithm, as stated in our article, can result in learned filters that are outside of the input domain, and thus would poorly represent the true cluster centers. Additionally, our learning rule does not require a softmax function to produce the neuron outputs.
>
> [1] G. Deco and D. Obradovic. "Decorrelated hebbian learning for clustering and function approximation". Neural Computation, 7(2):338–348, 1995. doi: 10.1162/neco.1995.7.2.338.

---

> > ### Comment · Reviewer_Vx3w · 2022-07-08
> > **Reply**
> >
> > Motivation:
> > The abstract of the current starts with:
> > "Despite the recent success of artificial neural networks, more biologically plausible learning methods may be needed to resolve the weaknesses of backpropagation trained models such as catastrophic forgetting and adversarial attacks."
> > Even if not directly claimed, one would expect that the paper is going to focus on a learning method, which will be more biologically plausible, and will address some issues such as catastrophic forgetting and adversarial attacks. The same idea re-appears later on in the introduction.
> > In my opinion, this is misleading, as as you mentioned, you make no claims on the biological plausibility of the suggested model, nor address catastrophic forgetting and adversarial attacks in any way. While the paragraph added at the end of the introduction is a step in the right direction, I still think that the motivation should be restructured and clarified.
> >
> > Writing:
> > It's unclear to me if you intend to divide the introduction into 2 parts or not.
> >
> > Experimental design:
> > From seeing the filters of the cluster alone, it is not convincing that the proposed algorithm learns anything meaningful. I have no intuition why the shown filters represent the data, especially as the same filters are achieved for different datasets as seen in subfigures 2f-g.
> > This could be resolved by comparing the proposed method to other clustering algorithms (to my understanding, any clustering algorithm will do), showing metrics for evaluation of the resulting clusters, so one could assess the quality of the presented learning rule.
> > Even a comparison to random filter assignments -- which is not very informative -- could strengthen the paper, as it will show that learning rule is learning something useful.

---

> > > ### Author Response · Authors · 2022-07-10
> > > **Response to reviewer Vx3w**
> > >
> > > Thank you again for your feedback. We have uploaded another updated version of the manuscript.
> > >
> > > > Motivation: The abstract of the current starts with: "Despite the recent success of artificial neural networks, more biologically plausible learning methods may be needed to resolve the weaknesses of backpropagation trained models such as catastrophic forgetting and adversarial attacks." Even if not directly claimed, one would expect that the paper is going to focus on a learning method, which will be more biologically plausible, and will address some issues such as catastrophic forgetting and adversarial attacks. The same idea re-appears later on in the introduction. In my opinion, this is misleading, as as you mentioned, you make no claims on the biological plausibility of the suggested model, nor address catastrophic forgetting and adversarial attacks in any way. While the paragraph added at the end of the introduction is a step in the right direction, I still think that the motivation should be restructured and clarified.
> > >
> > > The abstract now states that we do not specifically address the weaknesses of backpropagation trained models. Furthermore, the third and fourth paragraphs of the Introduction have been updated such that it is clearer that the proposed learning rule has many of the properties that a more biologically plausible method requires.
> > >
> > > > Writing: It's unclear to me if you intend to divide the introduction into 2 parts or not.
> > >
> > > We have now divided the Introduction into Introduction and Related Work.
> > >
> > > > Experimental design: From seeing the filters of the cluster alone, it is not convincing that the proposed algorithm learns anything meaningful. I have no intuition why the shown filters represent the data, especially as the same filters are achieved for different datasets as seen in subfigures 2f-g. This could be resolved by comparing the proposed method to other clustering algorithms (to my understanding, any clustering algorithm will do), showing metrics for evaluation of the resulting clusters, so one could assess the quality of the presented learning rule. Even a comparison to random filter assignments -- which is not very informative -- could strengthen the paper, as it will show that learning rule is learning something useful.
> > >
> > > An example result from k-means was added in Subfigure 2h, and we added a paragraph at the end of the MNIST section discussing this result.

---

### Author Response · Authors · 2022-08-10
**Link to complete article**

Complete article with author name and affiliation is available at https://arxiv.org/abs/2205.00920.

---

### Decision · Action_Editors · 2022-08-04

**Recommendation:** Reject

**Comment:**

This paper presents an online clustering algorithm based on Gaussian neurons based on a local learning rule. The reviewers noted a number of weaknesses in the submission, including unclear motivation, poor experimental design, inadequate comparisons to prior work, and unclear messaging. Some of these weaknesses were discussed and at least partially addressed in the authors' responses and in the updated manuscript. For example, the motivation and discussion of distribution shift and catastrophic forgetting was clarified so that the abstract and introduction more directly reflect the contributions of the paper. Still, a number of the reviewers' concerns persist after the revisions and discussion. For example, there are still concerns regarding the experimental design and comparisons to prior work, and the high-level takeaways are not clearly conveyed through the overall narrative nor adequately supported by evidence. As such, the reviewers and I are all in agreement that at present this paper is not suitable for publication at TMLR.